# Disease-Free and Overall Survival Implications of Pelvic Lymphadenectomy in Endometrial Cancer: A Retrospective Population-Based Single-Center Study

**DOI:** 10.3390/cancers15235636

**Published:** 2023-11-29

**Authors:** Marcin Misiek, Kaja Michalczyk, Aleksandra Kukla-Jakubowska, Marcin Lewandowski, Agnieszka Wrona-Cyranowska, Magdalena Koźmińska, Piotr Fras, Anita Chudecka-Głaz

**Affiliations:** 1Department of Gynecologic Oncology, Holy Cross Cancer Center, 25-734 Kielce, Poland; marcin.misiek@onkol.kielce.pl (M.M.); marcinle@onkol.kielce.pl (M.L.); agnieszkawr@onkol.kielce.pl (A.W.-C.); magdalenak@onkol.kielce.pl (M.K.); piotrfr@onkol.kielce.pl (P.F.); 2Department of Gynecological Surgery and Gynecological Oncology of Adults and Adolescents, Pomeranian Medical University, 70-204 Szczecin, Poland; aleksandra.kukla@pum.edu.pl (A.K.-J.); anitagl@poczta.onet.pl (A.C.-G.)

**Keywords:** endometrial cancer, lymph nodes, lymphadenectomy, overall survival, recurrence

## Abstract

**Simple Summary:**

Endometrial cancer is the most common gynecological malignancy, and its incidence is still on the rise. Surgical treatment is the most important step in patient treatment protocols and consists of hysterectomy, bilateral salpingo-oophorectomy and lymph node assessment. This retrospective single-center study analyzes the influence of pelvic lymphadenectomy on the disease-free and overall survival of endometrial cancer patients.

**Abstract:**

The role of pelvic lymphadenectomy in endometrial cancer remains unclear. In this study, we aimed to investigate the influence of lymphadenectomy on progression-free and overall survival among patients diagnosed with endometrial carcinoma. This retrospective single-center study included 1532 patients operated on in a Polish reference center for gynecologic oncology at Holy Cross Hospital, Kielce, between 2002 and 2020. A total of 1004 patients underwent systematic lymphadenectomy as a part of their surgical procedure. The median number of collected lymph nodes was seven. In total, 11.6% of patients were found to have lymph node invasion. The number of lymph nodes removed correlated with patient survival. In patients in whom the number of removed lymph nodes was above the median (>7), the risk of death was reduced (HR 0.68, *p* = 0.002). The risk of death correlated with the presence of lymph node metastasis (HR 4.12, *p* < 0.001). The risk of cancer progression was associated with the number of lymph nodes removed (HR 0.54, *p* = 0.006), and the risk of EC recurrence was greater in patients with lymph node metastasis (HR 1.94, *p* = 0.016). Our study provides additional evidence that systematic lymphadenectomy may influence the disease-free and overall survival of patients with endometrial cancer. The number of lymph nodes removed correlated with patient prognosis. Further studies are needed to evaluate the use of lymphadenectomy in endometrial cancer treatment.

## 1. Introduction

Endometrial cancer is the most common gynecological malignancy and remains one of the biggest challenges of gynecological oncology due to its constantly increasing prevalence. As the sizes of the aging and obese populations are rising, it is estimated that its prevalence will also continue to rise. Endometrial cancer is usually associated with a relatively good patient prognosis, with an approximately 76% overall patient survival. This is associated with early-stage patient diagnosis attributed to postmenopausal bleeding [1]. However, despite improving and minimally invasive surgical treatment and adjuvant treatment options, patients’ survival does not seem to be affected.

Surgical treatment is the most important element in endometrial cancer treatment. It consists of hysterectomy followed by bilateral salpingo-oophorectomy, which can either be performed laparoscopically or through laparotomy. However, the approach to lymph node staging remains controversial. The spread of endometrial cancer beyond the uterine tissue occurs through direct infiltration through the myometrium, further extending into the cervix, and the creation of cancer metastasis through the pelvic lymph nodes and, less frequently, the para-aortic lymph nodes. In the early stages of EC, when the disease is confined to the corpus uterus, lymph node metastases are found in approximately 10% of patients [2,3]. The percentage is lower among patients diagnosed with well-differentiated tumors and only superficial myometrial invasion; however, the rates increase up to 20% in female patients with poorly differentiated disease and deep myometrial invasion [2].

In this study, we aimed to investigate the influence of lymphadenectomy on patients’ PFS, OS and 5-year OS.

## 2. Materials and Methods

This was a retrospective single-center study conducted at the Department of Gynecologic Oncology, Holy Cross Cancer Center, Kielce, Poland. A total of 1532 patients diagnosed with EC and operated on at the department between 2002 and 2020 were included in the study. The study was conducted in accordance with the Declaration of Helsinki and approved by the Ethics Committee of Jan Kochanowski University of Kielce, Poland (protocol code 39/2023 and date of approval 8 September 2023). The data collected for the purpose of the study were gathered retrospectively based on hospital documentation and pathology reports.

Numerical variables are described using the mean and standard deviation or the median and interquartile range, depending on the distribution. Nominal traits are summarized using the absolute frequency and proportion. A distribution normality check was performed using the Shapiro–Wilk test and verified via skewness and kurtosis, while homogeneity was assessed using the Levene test. Comparisons were made using the Student *t* test for independent groups, Mann–Whitney U test, Pearson chi-square test or Fisher exact test, as appropriate. A correlation analysis was performed using the Spearman method. A survival analysis was run using Kaplan–Meier curves and the log-rank test for survival differences between groups. Overall survival was defined as the time from primary surgery to death from any cause. Progression-free survival was also calculated from the time of primary surgery to the first reappearance of endometrial cancer or death from any cause. Patients who were known to be alive and without any recurrent disease at the time of the study analysis were censored at the time of the last patient follow-up. Additionally, 5-year overall survival models were created. A proportional hazard Cox regression model was employed to quantify the impact of selected variables on survival. A regression analysis was run in 2 steps: a univariate analysis and a multivariate analysis. The final shape of the multivariate model was based on stepwise variable selection. The covariates included in the Cox regression model for overall and progression-free survival included tumor histology, the presence of vascular infiltration, tumor grading, FIGO staging, the conduction of lymphadenectomy, the number of lymph nodes removed, the number of lymph nodes affected and the site of the affected lymph nodes. The numbers of obtained lymph nodes and lymph node invasions were obtained from the pathology report. All statistical analyses were performed in R software, version R-4.1.2. An alpha of 0.05 was assumed to be statistically significant.

## 3. Results

The average age of the participants was 64.31 ± 9.99 years. Less than half were at the IA FIGO stage (43.2%), the IB stage was seen in 18.7% of cases, and 20.5% were diagnosed with FIGO II. The majority of surgeries were performed using laparotomy (72.1%). The mean procedure duration was 148.72 ± 51.41 min. The median blood loss was 200 mL, and blood transfusion took place in 4.3% of procedures. Reoperation was required in five patients. Relapse was observed in 6.7% of cases. Over one in five patients lived for at least 5 years (22.5%), and one in three died over the course of the analysis timeframe (30.3%). The median follow-up was 28.04 (15.45; 37.55) months. Systematic lymphadenectomy was conducted in 65.7% of patients. The detailed characteristics of the study population are presented in Table 1.

Out of the 1532 patients included in this study, 1004 patients underwent systematic lymphadenectomy as a part of the surgical procedure. The systematic lymphadenectomy consisted of the removal of all node tissue along the obturator fossa and the external iliac vessels up to the iliac bifurcation. The rest of the studied population either underwent sentinel node biopsy or did not undergo lymphadenectomy. The median number of collected lymph nodes was 7. A total of 11.6% of patients were found to have lymph node invasion (131 patients). The detailed results are presented below in Table 2.

The number of collected lymph nodes differed significantly between the type of surgical procedure (laparoscopy vs. laparotomy). In our study, in patients in whom pelvic lymphadenectomy was performed, significantly more lymph nodes were collected during laparoscopic hysterectomy (10) than during laparotomy (6) (*p* < 0.001). There was also a significant difference between the procedure duration, with a longer duration of laparotomies (159.26 min vs. 121.16 min, *p* < 0.001), and the need for blood transfusion (*p* = 0.003). However, the type of surgery did not influence the risk of reoperation (*p* = 0.626).

### 3.1. Comparison of Lymphadenectomy Parameters Depending on the Type of Surgical Procedure (Laparoscopy vs. Laparotomy)

Pelvic lymphadenectomy was conducted in 51.8% of patients who underwent total laparoscopic hysterectomy (TLH) and in 66.2% of patients who underwent laparotomy. Also, the frequency of paraaortic lymphadenectomy was higher when the THL procedure was conducted (31.6% vs. 19.2% in the case of laparotomy) (*p* = 0.002). Lymph node metastases were found in six patients within the TLH group (3.0%) and in 14.0% of patients in the laparotomy group (*p* < 0.001).

The number of removed lymph nodes above the median was more frequent among patients who underwent laparoscopic procedures than among those who underwent laparotomy (64.3% vs. 39.4%), with a significant difference (*p* < 0.001). The results are presented in Table 3.

### 3.2. Survival Analysis—Progression-Free Survival (PFS)

Our study found significant differences in progression-free survival (PFS) depending on the conduction of lymphadenectomy (*p* = 0.006). Patients who underwent lymphadenectomy had a higher PFS than the group of patients who did not undergo lymphadenectomy (39.30 months, CI_95_ [35.58; 43.02] vs. 29.74 months, CI_95_ [26.33; 33.16]). The median number of collected lymph nodes during the surgical procedure was 7. Patients in whom the number of collected lymph nodes was greater than the median were found to have a longer PFS than patients in whom the number of gathered lymph nodes was lower than or equal to the median (*p* = 0.006, M = 40.93, CI_95_ [35.55; 46.31] and M = 35.88, CI_95_ [32.03; 39.73], respectively).

Our results also confirmed the importance of endometrial cancer risk factors in patients’ PFS. Patients diagnosed with endometrioid carcinoma had a significantly longer PFS than patients diagnosed with other EC histologies (*p* < 0.001). Vascular infiltration, poor tumor differentiation grade (G3) and FIGO staging higher than FIGO IA were found to be poor patient prognostic factors and to negatively affect patients’ PFS (*p* < 0.001, *p* = 0.025, and *p* = 0.001, respectively). Also, patients diagnosed with type II EC using the Bokhman classification were found to have a shorter PFS than patients diagnosed with type I EC (*p* < 0.001). The specific results are presented in Table 4 and Figure 1.

#### Cox Model—Influence of Lymphadenectomy Parameters on PFS

After conducting the univariate proportional hazard Cox model, we found lymphadenectomy to be associated with a 42% lower risk of progression (HR = 0.58, CI_95_ [0.39;0.86], *p* = 0.006). In patients with any lymph node involvement, the risk of progression doubled (HR = 1.94, CI_95_ [1.13; 3.33], *p* = 0.016). The risk differed depending on the side of the affected iliac nodes: in patients with right lymph node invasion, the risk increased by ×3, and in the case of left pelvic lymph node invasion, the risk increased by ×2 (HR = 2.68, CI_95_ [1.44; 4.96], *p* = 0.002 and HR = 1.90, CI_95_ [1.02; 3.53], *p* = 0.042, respectively). When compared to patients without any lymph node metastasis, patients with right lymph node or left lymph node invasion had an increased risk of disease progression by ×3 and ×2 (HR = 2.61, CI_95_ [1.40; 4.85], *p* = 0.002 and HR = 1.92, CI_95_ [1.03; 3.57], *p* = 0.040, respectively). If both right and left lymph nodes were affected, the risk of progression doubled (HR = 1.86, CI_95_ [1.07; 3.23], *p* = 0.028).

The number of surgically collected lymph nodes during lymphadenectomy influenced patients’ PFS. If the number of lymph nodes was higher than 7 (median), the risk was 46% lower than when the number of lymph nodes was lower than or equal to the median (HR = 0.54, CI_95_ [0.35; 0.84], *p* = 0.006). If any of the collected lymph nodes were invaded, the risk of progression was ×2 higher than in patients with no lymph nodes affected (HR = 1.94, CI_95_ [1.13; 3.33], *p* = 0.016).

Vascular infiltration was associated with a nearly ×4 higher risk of progression (HR = 3.91, *p* < 0.001). Also, tumor grading G3 and higher FIGO staging were associated with an increased risk of disease progression (HR = 1.87, *p* = 0.028 and HR = 2.17, *p* = 0.002, respectively). A stepwise selection of variables was employed to create a multivariate proportional hazard model. Lymphadenectomy and endometrioid adenocarcinoma had a significant impact on the risk of progression. Having lymphadenectomy decreased the risk by 53% (HR = 0.47, CI_95_ [0.26; 0.83], *p* = 0.010). Endometrioid adenocarcinoma was associated with a 62% lower risk (HR = 0.38, CI_95_ [0.15; 1.00], *p* = 0.049). The detailed findings are presented in Table 5.

### 3.3. Survival Analysis—5-Year Overall Survival

The mean 5-year overall survival time for the whole study population was 32.61 months (CI_95_ [31.28; 33.93]). The mean 5-year OS was higher for patients who underwent lymphadenectomy than for patients who did not undergo this type of procedure (34.55 months, CI_95_ [32.86; 36.24] vs. 29.42, CI_95_ [27.30; 31.54], *p* = 0.005).

In our study, significant differences were observed between the groups of patients with lymph node invasion and no lymph node metastasis (*p* < 0.001), with a median survival of 28.62 months (CI_95_ [25.08; 32.16]) for patients with lymph node invasion vs. 34.96 months (CI_95_ [33.28; 36.63]) for patients with no nodal metastasis. If the lymph nodes were affected bilaterally, the median patient survival was 30.10 months (CI_95_ [26.43; 33.76]). The 5-year OS was influenced by the number of lymph nodes removed during the surgical procedure. Patients in whom the number of removed lymph nodes was greater than the median had a significantly better 5-year OS than patients in whom the number of removed lymph nodes was lower (35.44 months, CI_95_ [33.08; 37.81] vs. 32.87 months, CI_95_ [30.88; 34.85], *p* = 0.035). The 5-year OS was also influenced by the patients’ histopathological diagnosis, vascular infiltration status, tumor grading and FIGO staging. The detailed results are presented in Table 6 and Figure 2.

#### Cox Model—Influence of Lymphadenectomy Parameters on 5-Year Overall Survival

The univariate proportional hazard Cox model showed that lymphadenectomy decreased the risk of death by 30% within five years (HR = 0.70, CI_95_ [0.54; 0.90], *p* = 0.005). In patients in whom lymph node invasion was found, the risk of death within 5 years was increased by ×3 and ×4 (HR = 3.20, CI_95_ [2.33; 4.38], *p* < 0.001, and HR = 4.03, CI_95_ [2.13; 7.61], *p* < 0.001, respectively). A greater HR was found in patients in whom the right lymph nodes were affected than in patients with left lymph node invasion (HR = 3.57, CI_95_ [2.54; 5.02], *p* < 0.001 and HR = 2.36, CI_95_ [1.63; 3.41], *p* < 0.001, respectively). In patients with bilateral lymph node invasion, the risk of death within five years was 3x higher (HR = 2.91, CI_95_ [2.11; 4.01], *p* < 0.001). The hazard ratio was lower in patients with more lymph nodes collected during the surgical procedure (HR = 0.72, CI_95_ [0.53; 0.98], *p* = 0.036).

Patients diagnosed with endometrioid adenocarcinoma had a better prognosis than patients with other histopathological findings (HR = 0.41, CI_95_ [0.31; 0.55], *p* < 0.001). However, vascular infiltration, G3 tumor grading and FIGO staging greater than IA were found to be negative prognostic factors. The specific results are listed in Table 7.

A stepwise selection of variables was employed to create a multivariate proportional hazard model. Lymphadenectomy, a histopathological diagnosis of endometrioid adenocarcinoma, the presence of vascular infiltration and FIGO staging higher than IA were found to have a significant impact on the risk of death within five years. Lymphadenectomy decreased the risk by 48% (HR = 0.52, CI_95_ [0.31; 0.88], *p* = 0.015). Endometrioid adenocarcinoma of the HP type was associated with a 54% lower risk (HR = 0.46, CI_95_ [0.24; 0.90], *p* = 0.023). A doubled risk was observed in the case of both vascular infiltration and FIGO higher than IA (HR = 2.05, CI_95_ [1.14; 3.69], *p* = 0.017 and HR = 2.09, CI_95_ [1.09; 4.00], *p* = 0.026, respectively; Table 7).

### 3.4. Survival Analysis—Overall Survival (OS)

The mean survival time for the whole study population was over 3.5 years (M = 42.88, CI_95_ [40.17; 45.58], in months).

For patients who underwent lymphadenectomy, the OS time was significantly longer than for patients who did not undergo lymphadenectomy (46.66 months; CI_95_ [43.03; 50.28] vs. 36.82 months CI_95_ [32.89; 40.76], *p* = 0.001). We also found differences in patient survival based on the number of collected lymph nodes. If the number of collected lymph nodes was greater than the median (7), patients had a better OS than patients with fewer lymph nodes collected (*p* = 0.002).

The OS was significantly shorter in patients with lymph node invasion (*p* < 0.001) (M = 36.71, CI_95_ [29.59; 43.83] vs. M = 46.77, CI_95_ [43.20; 50.33]) than in patients with no lymph node metastases. In the case of bilateral lymph node invasion, patient survival differed significantly (38.79 months, CI_95_ [31.28; 46.29] vs. 46.27 months CI_95_ [42.65; 49.88], *p* < 0.001). The specific data are presented in Table 8 and Figure 3.

Similar to PFS, there were also significant differences in patient OS based on tumor histopathology, the presence of vascular infiltration, tumor grading and FIGO staging (see Figure 3).

#### Cox Model—Influence of Lymphadenectomy Parameters on OS

Based on a univariate proportional hazard Cox model, we found that lymphadenectomy was associated with a 29% lower risk of death (HR = 0.71, CI_95_ [0.58; 0.87], *p* = 0.001). If any lymph nodes were affected, the risk of death quadrupled (HR = 4.12, CI_95_ [2.24; 7.57], *p* < 0.001). When compared to patients without lymph node metastasis, right lymph node invasion or left lymph node invasion increased the risk by ×3 or ×2 (HR = 2.61, CI_95_ [1.92; 3.55], *p* < 0.001 and HR = 1.89, CI_95_ [1.38; 2.59], *p* < 0.001, respectively). If the lymph nodes were affected bilaterally, the risk of death doubled (HR = 2.11, CI_95_ [1.60; 2.78], *p* < 0.001).

If the number of collected lymph nodes was greater than 7 (median), the risk of death was 32% lower than in patients in whom fewer lymph nodes were collected during the surgery (HR = 0.68, CI_95_ [0.54; 0.87], *p* = 0.002). If any of the collected lymph nodes were positive, the risk of death was 2× higher than in females with no lymph node invasion (HR = 2.23, CI_95_ [1.70; 2.92], *p* < 0.001).

There were also differences based on tumor characteristics. The histopathological diagnosis of endometrioid adenocarcinoma was associated with a lower risk of death by 43% (HR = 0.57, CI_95_ [0.44; 0.73], *p* < 0.001) compared to other histological findings. Vascular infiltration tripled the risk of death (HR = 2.89, CI_95_ [2.30; 3.63], *p* < 0.001). Also, patients with G3 grading were 2× more likely to die (HR = 2.19, CI_95_ [1.70; 2.83], *p* < 0.001). When FIGO was higher than IA, the risk of death grew by 80% (HR = 1.80, CI_95_ [1.42; 2.28], *p* < 0.001). When FIGO was higher than IB, the risk of death increased by 86% (HR = 1.86, CI_95_ [1.52; 2.28], *p* < 0.001). Patients with type II cancer had a 78% increased risk of death (HR = 1.78, CI_95_ [1.37; 2.30], *p* < 0.001).

A stepwise selection of variables was employed to create a multivariate proportional hazard model. Lymphadenectomy, a histopathological diagnosis of endometrioid adenocarcinoma and vascular infiltration had significant impacts on the risk of death in the multivariate model. Lymphadenectomy decreased the risk of death by 35% (HR = 0.65, CI_95_ [0.44; 0.98], *p* = 0.041). Endometrioid adenocarcinoma histology was associated with a 68% lower risk (HR = 0.32, CI_95_ [0.18; 0.58], *p* < 0.001). A doubled risk was observed in patients with vascular infiltration (HR = 2.29, CI_95_ [1.37; 3.84], *p* = 0.001; Table 9).

## 4. Discussion

Endometrial cancer is the most commonly diagnosed gynecological cancer in developed countries. The majority of patients are diagnosed at an early stage when the disease is confined to the uterus [4]. Even though lymph node invasion is only found in approximately 10% of patients [2,3], surgical staging with lymphadenectomy can help define the risk of recurrence and provide further knowledge regarding adjuvant treatment for high-risk patients [5]. Our research revealed results similar to those of previous studies, as the proportion of patients with pelvic node metastases was found to be equal to 11.6%, and 11.2% of patients were found to have bilateral lymph node invasion.

Among endometrial cancers, based on the histopathological subtypes, the most common form of EC is endometrioid adenocarcinoma. In our study, 88.8% of the patient population was diagnosed with endometrioid adenocarcinoma, providing similar results. Other EC subtypes (such as adenosquamous, clear cell and serous carcinomas) usually tend to present at a more advanced FIGO staging and tend to have a poorer patient prognosis, as they are typically high-grade tumors (G3 grading) [6]. The results of our study revealed similar findings, as the histopathological diagnosis of endometrioid carcinoma was a good prognostic factor for patient PFS and OS. However, patients presenting with other EC subtypes and/or G3 tumor grading and FIGO staging greater than FIGO IA tended to have a poorer prognosis and a higher hazard ratio of adverse events.

The spread of endometrial cancer occurs through a direct invasion of the surrounding tissues and through lymphatic spread to the pelvic lymph nodes (including the external and common iliac lymph nodes and para-aortic lymph nodes). The distant metastases in patients with advanced forms of the disease usually occur through hematological spread. Lymphadenectomy can help select high-risk patients requiring further adjuvant treatment. Previous randomized controlled trials have shown no effect of adjuvant radiotherapy on overall patient survival in early-stage endometrial cancer (patients diagnosed with FIGO stage I without G3 disease or evidence of lymphovascular space invasion). However, it was found to reduce the number of pelvic recurrences [7].

Lymphadenectomy seems to play an important role in the assessment of lymph node metastatic involvement. However, there is no proper definition of an “adequate” lymph node dissection or of the number of lymph nodes required to be removed for appropriate patient staging and the determination of lymph-node-negative disease. An analysis by Chan et al. showed that a higher number of recovered lymph nodes is associated with a greater chance of detecting at least one positive lymph node metastasis [8]. Their study also included an analysis of 11,443 patients in the National Cancer Registry, showing that removing 21–25 lymph nodes increases the probability of detecting at least 1 positive lymph node by 45% compared with removing only 1–5 lymph nodes.

There are conflicting results on the benefit of lymphadenectomy in patients diagnosed with EC cancer, and the therapeutic role of lymphadenectomy remains one of the greatest topics of debate in gynecologic oncology. Some studies have shown an improvement in patient survival [9,10,11], while others have questioned its therapeutic value [3,12]. A study by Chan et al. [11] evaluated the role of lymphadenectomy by reviewing 12,333 endometrioid EC patients. The authors found intermediate-/high-risk patients who underwent an extensive lymph node resection to have improved 5-year disease-specific survival. However, they did not show any survival advantages for low-risk cancer patients (defined as FIGO IA, all grades, and stage IB grades 1–2). The type of lymphadenectomy (pelvic (PLND) vs. pelvic and para-aortic (PPaLND)) also seems to influence patient survival. A retrospective cohort SEPAL study [13], as well as a meta-analysis study by Guo et al., showed patients at an intermediate- or high-risk of recurrence who underwent PPaLND to have better survival outcomes than those who underwent PLND only, especially with regard to OS [14]. However, a systematic review and meta-analyses based on randomized studies revealed no differences in the risk of recurrence or death between lymphadenectomy and standard surgery without lymph node removal [3,6,15]. There were also no differences in direct surgical morbidity; however, more patients were found to experience surgery-related systemic morbidity, lymphedema or lymphocysts [6]. Despite the prospective characteristics of the studies, they included a high number of patients, who, postoperatively, were found to have a low risk of EC recurrence and a low risk of lymph node metastasis. The median number of resected lymph nodes also varied between the studies, with a median of 12 removed lymph nodes in the ASTEC study [3], which might have affected patient survival.

This research was conducted on retrospective data based on surgeries conducted between 2002 and 2020. In this study, pelvic lymphadenectomy was conducted in 51.8% of patients who underwent total laparoscopic hysterectomy (TLH) and in 66.2% of patients who underwent laparotomy. The rest of the patients either underwent an SLN biopsy or did not undergo lymphadenectomy based on their medical condition, histopathological report, medical imaging and surgeons’ decision. The management of endometrial cancer treatment, especially regarding LN staging, has been debatable for a long time and has remained heterogeneous between different institutions. Historically, hysterectomy with systematic lymphadenectomy was the treatment of choice for EC patients [16,17]; however, no clinical trials have confirmed its benefit over SLN sampling [3,16,18]. In 2023, a new FIGO classification was introduced, dividing EC patients into risk categories based on histological type, tumor pattern and molecular classification [19]. Multiple trials, e.g., the PORTEC trial, have investigated the use of molecular classification to better stratify patients’ treatment and prognosis [20,21]. The classification accounted for specific histological and molecular futures based on diagnostic algorithms using p53, MSH6, PMS2 and POLE mutations to identify EC prognostic groups. Five categories of EC tumors were recognized, namely, ultramutated/pathogenic POLE mutations, hypermutated MSI/MMRd, high copy number/p53 abnormal status, low copy number/NSMP and a multiple classifier with any combination of the markers included in the previous categories. Numerous studies have found the use of this approach to have prognostic relevance, especially for determining high-risk tumors [20,22]. Incorporating risk stratification can help better define prognostic and therapeutic approaches to EC treatment evaluation in patients in whom, e.g., systematic lymphadenectomy can be beneficial. As this study was performed in the years 2002–2020, especially at the beginning of the study, the use of molecular classification to better stratify patients into risk group categories was not available. All of the patients in whom lymphadenectomy was technically possible underwent the procedure. In the years 2002–2010, all patients in the study underwent laparotomy. Since 2011, an increasing number of patients included in the study underwent laparoscopic treatment (from 2.4% of patients in 2011 up to 59% in 2020). With the increasing use of laparoscopy in endometrial cancer treatment in our center, the use of SLN biopsy has become more widely available, limiting the rates of total pelvic lymphadenectomy. The surgical treatment method (laparoscopy/laparotomy) and the extent of the surgery (the extent of the lymphadenectomy/SLN procedure) were decided by a team of doctors (the operator and the director of the department) based on patient characteristics (obesity, comorbidities and a previous history of abdominal surgeries), medical imaging studies and the histopathological and molecular characteristics of the tumor.

Our study found that lymphadenectomy influences patient survival (HR 0.71, *p* = 0.001 for OS). The number of lymph nodes removed during lymphadenectomy correlated with patient survival. In patients in whom the number of removed lymph nodes was above the median (>7), the risk of death was reduced (HR 0.68, *p* = 0.002). The risk of death correlated with the presence of lymph node metastasis (HR 4.12, *p* < 0.001). Similar findings were found regarding the risk of EC progression. Lymphadenectomy reduced the risk of cancer progression (HR = 0.58, *p* = 0.006) and was associated with the number of lymph nodes removed (HR 0.54, *p* = 0.006). The risk of EC recurrence was greater in patients with lymph node metastasis (HR 1.94, *p* = 0.016).

Also, the use of a minimally invasive approach (laparoscopy/robotic procedure) should be evaluated in comparison to laparotomy. In our study, 335 patients underwent total laparoscopic hysterectomy, while 867 patients underwent surgical laparotomy. We found the number of collected lymph nodes to differ significantly between the type of surgical procedure (laparoscopy vs. laparotomy), with significantly more lymph nodes being collected during the TLH procedure (10) than during laparotomy (6) (*p* < 0.001). This may be due to a better visualization of lymph nodes and more precise tissue dissection during laparoscopic procedures. However, in our study, patients who underwent laparotomy were more frequently diagnosed with lymph node metastasis. This may be caused by patient selection criteria, as patients with more advanced stages of endometrial cancer might have been more frequently qualified for laparotomy. Our study also shows significant differences in the length of the procedure, as the length of the surgery duration was longer among patients who underwent laparotomies. Such patients had a higher risk of blood transfusion. Previous studies have also investigated the role of a minimally invasive approach in the surgical treatment of EC patients. The results of the LAP2 study conducted by the GOG group showed a similar risk of intraoperative complications between the two groups, with a higher risk of postoperative complications among patients who underwent laparotomy. The rates of pelvic lymph node dissection were similar regardless of the method of surgical treatment.

There are no international guidelines for “adequate” systematic lymphadenectomy regarding node counts. The number of lymph nodes removed during lymphadenectomy depends on multiple factors, including patient characteristics (obesity, a history of previous abdominal surgeries, the presence of adhesions and immunologic status), surgical thoroughness, surgeons’ skills and the experience of the cancer center, as well as the pathologist and pathological examination of tissues. Visualization and palpation, fat clearing and entire submission are the standard techniques used for lymph node assessment [23]. When using visualization and palpation, smaller LNs can be missed, while fat clearing requires intensive effort and may not provide relevant information [24].

The extent of lymphadenectomy may influence treatment-related morbidity and mortality. Moreover, it can prolong the duration of the surgical procedure and elevate treatment costs [3,5]. The perioperative complications include infections, the need for blood transfusions, lymphocyst formation, leg edema, deep vein thrombosis and bowel obstruction and may be present in up to 20% of patients [25]. The complication rate is also higher in patients who undergo radiation treatment after lymphadenectomy [26,27]. The ASTEC randomized clinical trial showed a substantial increase in lymphedema incidence among patients who underwent lymphadenectomy compared to those who underwent standard surgery [3]. The ASTEC trial is one of the most extensive randomized trials on systematic lymphadenectomy. The study showed no improvement in 5-year progression-free survival from lymphadenectomy. Based on the study results, a surgeon aiming for systematic lymphadenectomy should consider the balance of risks and benefits associated with the procedure. Even though the results of systematic trials suggest that there are no therapeutic effects of lymphadenectomy, lymphadenectomy can be used for surgical staging and patient stratification to identify the need for adjuvant treatment (chemo- or radiotherapy).

An ESGO/ESTRO/ESP consensus and guidelines for the management of patients with endometrial carcinoma were created in 2020, updating the guidelines on EC treatment [28]. The guidelines summarize the standards of patient treatment, including lymph node staging. In accordance with the guidelines, a sentinel node biopsy can be used as an alternative to lymph node dissection for lymph node staging. Numerous studies, including cohort prospective studies, have confirmed the use of sentinel lymph nodes for endometrial cancer lymph node staging and revealed a high sensitivity of this method in patients with early-stage endometrial carcinoma; e.g., the FIRES trial [29] investigated the use of lymph node biopsy and lymphadenectomy in endometrial cancer staging. The authors found a high diagnostic accuracy in the detection of EC metastasis. Only in approximately 3% of patients with node-positive disease was the sentinel lymph node biopsy unable to identify node-positive diseases. The use of a sentinel lymph node biopsy instead of pelvic lymphadenectomy carries multiple advantages, lowering the risk of patient postoperative morbidity, including postsurgical complications, such as lower leg edema, lymphocyst formation and injury to the genitofemoral nerve [30,31]. Moreover, lymphadenectomy is a technically difficult procedure, especially in an obese and older population, which represents a substantial number of patients diagnosed with endometrial cancer. Recently, some retrospective studies demonstrated that the prognosis of patients who receive a complete lymphadenectomy is similar to that of patients who receive a sentinel biopsy only. However, there are still limited data regarding patient oncologic outcomes, especially for high-risk patients. Numerous studies assessed its safety and accuracy in evaluating the nodal status of early EC, and it has become the standard of care in many oncological centers [32], even though there are still some discrepancies between the international guidelines on SLN mapping.

What needs to be noted is that lymphadenectomy can not only be used for patient staging but also as a therapeutic procedure, as it involves the removal of involved lymph nodes, which can be potential sites of future pelvic recurrences [6]. The role of lymphadenectomy in endometrial cancer remains unknown, and the excision of pelvic lymph nodes may not directly provide a therapeutic benefit but may allow for patient stratification into prognostic groups.

## 5. Conclusions

As the role of lymphadenectomy in the surgical management of endometrial cancer remains controversial, further studies are needed to evaluate the use of lymphadenectomy in endometrial cancer treatment. Based on the current evidence and knowledge, future studies should aim to investigate the specific groups of patients at risk of lymph node metastasis formation, their selection, the role of different diagnostic techniques (including SLN mapping) and specific patient treatment as well as its consequences on patient morbidity, patient quality of life and costs of treatment.

## Figures and Tables

**Figure 1 cancers-15-05636-f001:**
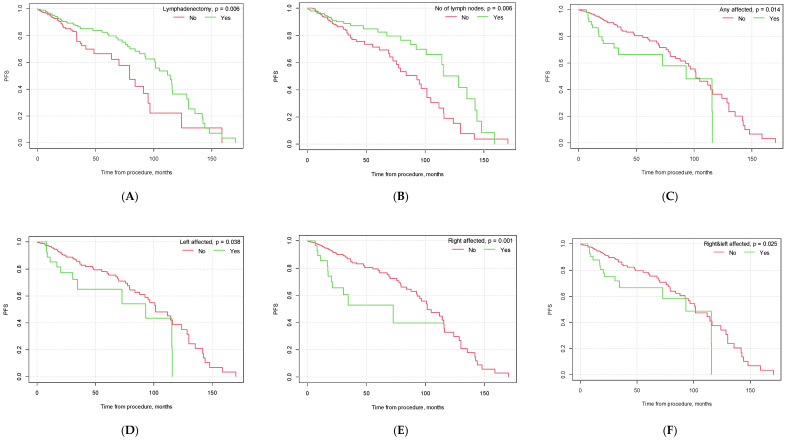
Kaplan–Meier progression-free survival curves depending on selected factors ((**A**) lymphadenectomy vs. no lymphadenectomy; (**B**) no. of lymph nodes removed below vs. above median; (**C**) pelvic lymph node metastasis yes vs. no; (**D**) metastasis in left pelvic lymph nodes yes vs. no; (**E**) metastasis in right pelvic lymph nodes yes vs. no; (**F**) metastasis in both left and right pelvic lymph nodes yes vs. no; (**G**) G3 grading vs. other; (**H**) type II EC vs. other; (**I**) endometrioid carcinoma vs. other histological types of EC; (**J**) vascular infiltration yes vs. no; (**K**) FIGO > 1A yes vs. no; (**L**) FIGO > IB yes vs. no).

**Figure 2 cancers-15-05636-f002:**
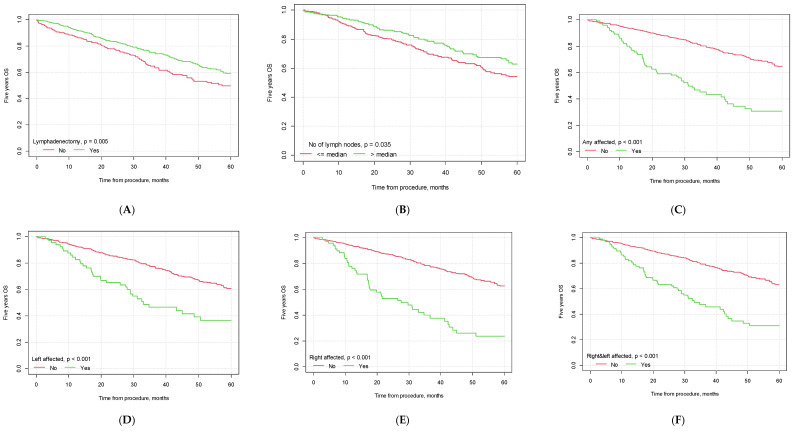
Kaplan–Meier 5-year overall survival curves depending on selected factors ((**A**) lymphadenectomy vs. no lymphadenectomy; (**B**) no. of lymph nodes removed below vs. above median; (**C**) pelvic lymph node metastasis yes vs. no; (**D**) metastasis in left pelvic lymph nodes yes vs. no; (**E**) metastasis in right pelvic lymph nodes yes vs. no; (**F**) metastasis in both left and right pelvic lymph nodes yes vs. no; (**G**) G3 grading vs. other; (**H**) type II EC vs. other; (**I**) endometrioid carcinoma vs. other histological types of EC; (**J**) vascular infiltration yes vs. no; (**K**) FIGO> 1A yes vs. no; (**L**) FIGO > IB yes vs. no).

**Figure 3 cancers-15-05636-f003:**
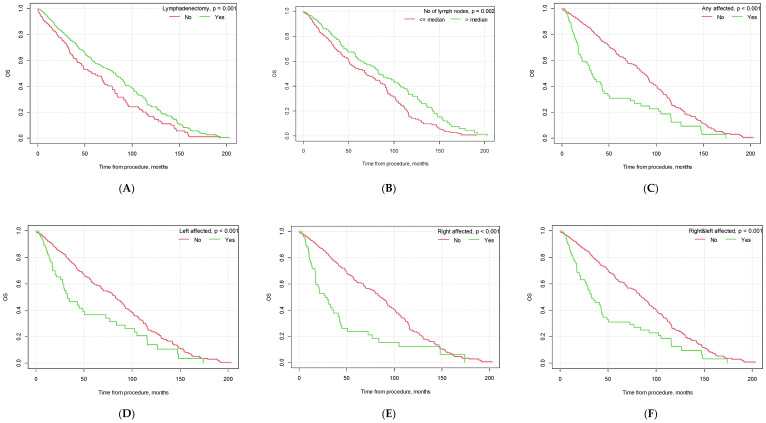
Kaplan–Meier overall survival curves depending on selected factors ((**A**) lymphadenectomy vs. no lymphadenectomy; (**B**) no. of lymph nodes removed below vs. above median; (**C**) pelvic lymph node metastasis yes vs. no; (**D**) metastasis in left pelvic lymph nodes yes vs. no; (**E**) metastasis in right pelvic lymph nodes yes vs. no; (**F**) metastasis in both left and right pelvic lymph nodes yes vs. no; (**G**) G3 grading vs. other; (**H**) type II EC vs. other; (**I**) endometrioid adenocarcinoma vs. other histological types of EC; (**J**) vascular infiltration yes vs. no; (**K**) FIGO > 1A yes vs. no; (**L**) FIGO > IB yes vs. no).

**Table 1 cancers-15-05636-t001:** Study group characteristics.

Variable	*n* (%)
Age (years) mean ± SD	64.31 ± 9.99
FIGO	
IA	653 (43.2)
IB	282 (18.7)
II	310 (20.5)
IIIA	62 (4.1)
IIIB	46 (3.0)
IIIC	3 (0.2)
IIIC1	94 (6.2)
IIIC2	15 (1.0)
IVA	5 (0.3)
IVB	41 (2.7)
Grading	
G1	754 (50.8)
G2	517 (34.8)
G3	198 (13.3)
Gx	16 (1.1)
Histopathology	
Endometrioid carcinoma	1352 (88.8)
Clear cell carcinoma	31 (2.0)
Mixed carcinoma	29 (1.9)
Papillary carcinoma	57 (3.7)
Other	54 (3.5)
Bokhman Classification	
Type I	1369 (89.5)
Type II	147 (9.6)
Vascular infiltration	
Yes	266 (17.5)
No	1254 (82.5)
Procedure timing	
2002	28 (1.8)
2003–2008	364 (23.8)
2009–2014	530 (34.6)
2015–2020	610 (39.8)
Procedure type	
Total laparoscopic hysterectomy	335 (27.9)
Laparotomy	867 (72.1)
Procedure duration, min, mean ± SD	148.72 ± 51.41
Blood loss, mL, median (IQR)	200.00 (180.00; 300.00)
Hospital stay after the procedure, days, median (IQR)	6 (5; 8)
Blood transfusion	24 (4.3)
Reoperation	5 (1.4)
Relapse	37 (6.7)
5-year survival	183 (22.5)
Death during the follow-up period	452 (30.3)

SD—standard deviation, IQR—interquartile range.

**Table 2 cancers-15-05636-t002:** Lymphadenectomy characteristics.

	N of Patients (%)
Lymphadenectomy	1004 (65.7)
No. of lymph nodes, median (IQR)	7 (3; 12)
No. of right lymph nodes, median (IQR)	3 (1; 6)
No. of left lymph nodes, median (IQR)	3 (0; 6)
Any lymph nodes affected	131 (11.6)
Right lymph nodes affected	92 (8.3)
Left lymph nodes affected	80 (7.3)
Bilateral lymph node involvement	122 (11.2)

IQR—interquartile range.

**Table 3 cancers-15-05636-t003:** Comparison of parameters related to lymphadenectomy depending on type of procedure.

Variable	THL (*n* = 335)	Laparotomy (*n* = 867)	*p*
No. of patients who underwent pelvic lymphadenectomy	173 (51.8)	573 (66.2)	<0.001
No. of patients who underwent paraaortic lymphadenectomy	37 (31.6)	63 (18.2)	0.002
No. of patients with any pelvic lymph nodes affected	6 (3.0)	91 (14.0)	<0.001
No. of patients with any aortic lymph nodes affected	2 (1.8)	14 (4.2)	0.543 ^1^
No. of patients with a no. of collected lymph nodes above median	128 (64.3)	256 (39.4)	<0.001
No. of patients with a no. of lymph nodes affected above median	6 (3.0)	91 (14.0)	<0.001

^1^ Data presented as *n* (%). Comparisons were made with Pearson chi-square test or Fisher exact test.

**Table 4 cancers-15-05636-t004:** Progression-free survival (PFS) based on patients’ characteristics.

	Median (Months), [95% CI]	*p*-Value
Histopathology		<0.001
Endometrioid	36.37, [33.47; 39.26]	
Other than endometrioid	30.40, [23.25; 37.55]	
Vascular infiltration		<0.001
Yes	24.24, [21.03; 27.44]	
No	38.77, [35.56; 41.99]	
Grading (G3 vs. other)		0.025
G3	30.04, [23.71; 36.36]	
Other than G3	36.76, [33.80; 39.71	
FIGO (higher than IA vs. IA)		0.001
Higher than IA	38.62, [34.80; 42.44]	
IA	32.34, [28.67; 36.01]	
FIGO (higher than IB vs. IA/IB)		0.001
Higher than IB	35.05, [30.96; 39.14]	
IA/IB	36.63, [33.02; 40.24]	
Bokhman classification		<0.001
Type II	30.76, [22.71; 38.81]	
Type I	36.26, [33.42; 39.11]	

**Table 5 cancers-15-05636-t005:** Proportional hazard Cox model outcomes for progression-free survival.

Variable	Univariate Models			Multivariate Model		
	HR	95% CI	*p*	HR	95% CI	*p*
Lymphadenectomy	0.58	0.39 to 0.86	0.006	0.47	0.26 to 0.83	0.010
Any lymph nodes affected	1.94	1.13 to 3.33	0.016	-	-	-
Right lymph nodes affected	2.68	1.44 to 4.96	0.002	-	-	-
Left lymph nodes affected	1.90	1.02 to 3.53	0.042	-	-	-
Right lymph nodes affected (vs. right and left not affected)	2.61	1.40 to 4.85	0.002	-	-	-
Left lymph nodes affected (vs. right and left not affected)	1.92	1.03 to 3.57	0.040	-	-	-
Right and left lymph nodes affected	1.86	1.07 to 3.23	0.028	-	-	-
No. of lymph nodes, above median	0.54	0.35 to 0.84	0.006	-	-	-
No. of lymph nodes affected, above median	1.94	1.13 to 3.33	0.016	-	-	-
HP endometrioid adenocarcinoma	0.42	0.26 to 0.67	<0.001	0.38	0.15 to 1.00	0.049
Vascular infiltration	3.91	2.44 to 6.27	<0.001	2.03	0.92 to 4.48	0.080
Grading G3	1.87	1.07 to 3.27	0.028	-	-	-
FIGO above IA	2.17	1.34 to 3.52	0.002	-	-	-
FIGO above IB	1.97	1.33 to 2.91	0.001	-	-	-
Type II EC	2.40	1.47 to 3.92	<0.001	-	-	-

HR—hazard ratio, CI—confidence interval.

**Table 6 cancers-15-05636-t006:** Five-year OS depending on patients’ characteristics.

	Median (Months), [95% CI]	*p*-Value
Histopathology		<0.001
Endometrioid	33.27, [31.83; 34.71]	
Other than endometrioid	28.51, [25.06; 31.96]	
Vascular infiltration		<0.001
Yes	24.57, [22.13; 27.01]	
No	35.16, [33.64; 36.68]	
Grading (G3 vs. other)		<0.001
G3	27.38, [23.99; 30.77]	
Other than G3	33.79, [32.32; 35.25]	
FIGO (higher than IA vs. IA)		<0.001
Higher than IA	32.83, [31.14; 34.53	
IA	32.33, [30.15; 34.52]	
FIGO (higher than IB vs. IA/IB)		<0.001
Higher than IB	30.26, [28.30; 32.22	
IA/IB	34.58, [32.77; 36.39]	
Bokhman classification		<0.001
Type II	28.12, [24.38; 31.86]	
Type I	33.27, [31.86; 34.69]	

**Table 7 cancers-15-05636-t007:** Proportional hazard Cox model outcomes for 5-year overall survival.

Variable	Univariate Models			Multivariate Model		
	HR	95% CI	*p*	HR	95% CI	*p*
Lymphadenectomy	0.70	0.54 to 0.90	0.005	0.52	0.31 to 0.88	0.015
Any lymph nodes affected	3.20	2.33 to 4.38	<0.001	-	-	-
Right lymph nodes affected	3.57	2.54 to 5.02	<0.001	-	-	-
Left lymph nodes affected	2.36	1.63 to 3.41	<0.001	-	-	-
Right lymph nodes affected (vs. right and left not affected)	3.68	2.61 to 5.21	<0.001	-	-	-
Left lymph nodes affected (vs. right and left not affected)	2.62	1.80 to 3.81	<0.001	-	-	-
Right and left lymph nodes affected	2.91	2.11 to 4.01	<0.001	-	-	-
No. of lymph nodes, above median	0.72	0.53 to 0.98	0.036	-	-	-
No. of lymph nodes affected, above median	3.20	2.33 to 4.38	<0.001	-	-	-
HP endometrioid adenocarcinoma	0.41	0.31 to 0.55	<0.001	0.46	0.24 to 0.90	0.023
Vascular infiltration	3.55	2.74 to 4.60	<0.001	2.05	1.14 to 3.69	0.017
Grading G3	2.63	1.98 to 3.50	<0.001	-	-	-
FIGO above IA	2.42	1.75 to 3.34	<0.001	2.09	1.09 to 4.00	0.026
FIGO above IB	2.62	2.01 to 3.41	<0.001	-	-	-
Type II EC	2.37	1.77 to 3.19	<0.001	-	-	-

HR—hazard ratio, CI—confidence interval.

**Table 8 cancers-15-05636-t008:** The overall survival (OS) depending on patients’ characteristics.

	Median (Months), [95% CI]	*p*-Value
Histopathology		<0.001
Endometrioid	43.94, [41.00; 46.89]	
Other than endometrioid	36.19, [29.37; 43.01]	
Vascular infiltration		<0.001
Yes	27.34, [23.52; 31.17]	
No	47.81, [44.55; 51.08]	
Grading (G3 vs. other)		<0.001
G3	31.95, [26.45; 37.44	
Other than G3	44.91, [41.87; 47.96]	
FIGO (higher than IA vs. IA)		<0.001
Higher than IA	43.34, [39.93; 46.75]	
IA	42.27, [37.71; 46.83]	
FIGO (higher than IB vs. IA/IB)		<0.001
Higher than IB	38.36, [34.59; 42.13]	
IA/IB	46.66, [42.81; 50.51]	
Bokhman classification		<0.001
Type II	35.98, [28.61; 43.36]	
Type I	43.90, [41.00; 46.81]	

**Table 9 cancers-15-05636-t009:** Proportional hazard Cox model outcomes for overall survival.

Variable	Univariate Models			Multivariate Model		
	HR	95% CI	*p*	HR	95% CI	*p*
Lymphadenectomy	0.71	0.58 to 0.87	0.001	0.65	0.44 to 0.98	0.041
Any lymph nodes affected	2.23	1.70 to 2.92	<0.001	-	-	-
Right and left lymph nodes affected	2.11	1.60 to 2.78	<0.001	-	-	-
No. of lymph nodes, above median	0.68	0.54 to 0.87	0.002	-	-	-
No. of lymph nodes affected	2.23	1.70 to 2.92	<0.001	-	-	-
HP endometrioid adenocarcinoma	0.57	0.44 to 0.73	<0.001	0.32	0.18 to 0.58	<0.001
Vascular infiltration	2.89	2.30 to 3.63	<0.001	2.29	1.37 to 3.84	0.002
Grading G3	2.19	1.70 to 2.83	<0.001	-	-	-
FIGO above IA	1.80	1.42 to 2.28	<0.001	-	-	-
FIGO above IB	1.86	1.52 to 2.28	<0.001	-	-	-
Type II EC	1.78	1.37 to 2.30	<0.001	-	-	-

HR—hazard ratio, CI—confidence interval.

## Data Availability

Data are available upon request.

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
