# Peer review of "Disease-Free and Overall Survival Implications of Pelvic Lymphadenectomy in Endometrial Cancer: A Retrospective Population-Based Single-Center Study"

_cancers, 2023, doi:10.3390/cancers15235636_

Round 1

Reviewer 1 Report

Comments and Suggestions for Authors

Thanks for kind invitation to review a manuscript entitled “Disease-free and overall survival implications of pelvic lymphadenectomy in endometrial cancer: a retrospective population-based single center study”.

This manuscript analyzed several important issues associated with lymphadenectomy in endometrial cancer.

My opinions are as follows:

I. Major

1) In this study, systematic LA seemed to be routinely performed in all endometrial cancer patients. Previously, a full standard lymphadenectomy (LA) was recommended for all patients. However, no randomized trial data support routine full lymphadenectomy. Many clinicians favor lymph node sampling or sentinel lymph node mapping instead of systematic LA in certain conditions. So, I think that there is significant consensus that systematic LA has limited therapeutic value in low risk group.

In 2023, FIGO revised staging system based on pathologic and molecular risk factors. It defined high grade endometrioid, serous, clear cell, mixed, undifferentiated, carcinosarcoma, mesonephric-like and gastrointestinal mucinous type as aggressive histological types. Although the staging system does not guide surgical strategies, systematic LA would be warranted in these high risk subtypes.

How about analyzing the role of systematic LA in these high risk groups?

2) Each clinician might have different criteria for the extent of LA. The median number of LNs retrieved seems to be low, if you performed systematic LA. Therefore, you would be better to mention about the extent of LA.

II. Minor

To improve readers’ understanding, this manuscript need for English and medical editing.

eg)

-       Line 46-58: description unrelated to the thesis

-       Materials and method: only statistical analyses were described. Please move ‘line 101-107 in results’ to this section

-       Subtypes of Histopathology in table 1; not English (Clarocellulare, etc)

-       There are some ambiguous descriptions: ex) one patient in five had FIGO II (20.5%) à (Among 1532) 310 (20.5%) had FIGO stage II. Etc

-       Figure 1/2 legends do not match with figures: please correct!

-       Line 161-162: Patients with lymph nodes invasion were found to have significantly higher PFS than patients in whom no lymph node invasion was detected (p=0.014): correct description? (higher PFS à worse PFS)

-       Line 174; endometrial adenocarcinoma à endometrioid adenocarcinoma

-       etc

Author Response

Dear reviewer, we would like to thank you for your comments

Please see our responses: 

1) In this study, systematic LA seemed to be routinely performed in all endometrial cancer patients. Previously, a full standard lymphadenectomy (LA) was recommended for all patients. However, no randomized trial data support routine full lymphadenectomy. Many clinicians favor lymph node sampling or sentinel lymph node mapping instead of systematic LA in certain conditions. So, I think that there is significant consensus that systematic LA has limited therapeutic value in low risk group.

Thank you for your comment. This research was conducted on retrospective data based on the surgeries conducted between 2002 and 2020. Out of 1532 patients who participated in the study, systematic pelvic lymphadenectomy was conducted in 1004 patients. Unfortunately, we did not analyze the patients by year and we have no data with regard to why in this group of patients the lymphadenectomy was conducted and in which the SLN procedure was conducted. In 335 patients laparoscopy (total laparoscopic hysterectomy) was conducted and in 51.6% of patients, systematic pelvic lymphadenectomy was conducted. Our data show a trend towards increasing use of SLN instead of systematic lymphadenectomy in EC patients, however, we did not analyze this dataWe have provided some additional explanation in the manuscript. Please see the improved version.

In 2023, FIGO revised staging system based on pathologic and molecular risk factors. It defined high grade endometrioid, serous, clear cell, mixed, undifferentiated, carcinosarcoma, mesonephric-like and gastrointestinal mucinous type as aggressive histological types. Although the staging system does not guide surgical strategies, systematic LA would be warranted in these high risk subtypes.How about analyzing the role of systematic LA in these high risk groups?

Thank you for your comment, as it is very interesting. In our study we had 31 clear cell, and 29 mixed carcinomas. 54 patients were classified as other, due to limited patient population. Multiple patients were qualified for surgery based on biopsy results and/or hysteroscopy/ ACU findings with limited histopathological findings that not always differentiated between specific cancer subtypes or the diagnosis was changed after the final histopathological examination

We would like to perform a subanalysis as a future study – most likely including newer data on patients recently treated or by creating a prospective study. It would be also interesting to account molecular classification with regard to patients PFS and OS evaluation

2) Each clinician might have different criteria for the extent of LA. The median number of LNs retrieved seems to be low, if you performed systematic LA. Therefore, you would be better to mention about the extent of LA.

For the purpose of the analysis, only patients who underwent systhematic lymphadenectomy were analyzed. All of the patients were operated by the specialists working at the institution. It is true, that the extent of LA may be surgeon depended, however there are also different criteria including patient characteristics and patomorphologists etc. We discuss the number of removed lymph nodes in lines 434-436, 452-458 and 466-483.

We added additional information regarding the extent of LA – lines 126 to 132

 Our data shows an increasing number of collected lymph nodes from the beginning of the study (2002) until recently. However, we have not conducted any specific statistical analysis comparing the number of removed lymphnodes in different years as this was not the purpose of the study. We would certainly like to concentrate on it in the future researches

  1. Minor

-       Line 46-58: description unrelated to the thesis – we deleted this fragment. We have used some of It in the discussion part in order to discuss the use of lymphadenectomy/ SLN

-       Materials and method: only statistical analyses were described. Please move ‘line 101-107 in results’ to this section – thank you for your suggestion, we moved this paragraph to methods

-       Subtypes of Histopathology in table 1; not English (Clarocellulare, etc) – please see the improved version of the table

-       There are some ambiguous descriptions: ex) one patient in five had FIGO II (20.5%) à(Among 1532) 310 (20.5%) had FIGO stage II. Etc – we have improved that

-       Figure 1/2 legends do not match with figures: please correct! – we reordered the figures, thank you for noticing

-       Line 161-162: Patients with lymph nodes invasion were found to have significantly higher PFS than patients in whom no lymph node invasion was detected (p=0.014): correct description? (higher PFS à worse PFS) – we have improved it thank you for noticing

-       Line 174; endometrial adenocarcinoma à endometrioid adenocarcinoma – we have changed that, sorry

Also, as suggested by the second reviewer, we added tables with the description of PFS and OS based on patients’ characteristics

Please see the improved version of the manuscript

Reviewer 2 Report

Comments and Suggestions for Authors

In results, the order of figure 1 and 2 might be reversed.

And the duration(months) of PFS and OS is not described by table at all.

In my opinion, all results with number must be described by table.

What means affected side dependent results' differene?

If you have any evidence or reference looks like that, please show me the information.

Comments on the Quality of English Language

Minor editing is necessary including spellings, spacing rules.

'Recurrencce free survival' might be changed and united into 'Progression free survival'.

Author Response

Dear reviewer,

We would like to thank you for your comments

Please see the improved version of the manuscript and responses to your comments

In results, the order of figure 1 and 2 might be reversed. - we reordered the figures, thank you for noticing

And the duration(months) of PFS and OS is not described by table at all. In my opinion, all results with number must be described by table. – we added three additional tables describing the PFS and OS based on patients’ characteristics. Please see the improved version of the manuscript

What means affected side dependent results' differene?If you have any evidence or reference looks like that, please show me the information. – we have deleted this part of the article as we did not find any appropriate bibliography

Comments on the Quality of English Language

Minor editing is necessary including spellings, spacing rules. – thank you for your comment, we tried to edit the editorial errors. Moreover, we would like to ask the editorial office to help with the text outlook prior to publication

'Recurrencce free survival' might be changed and united into 'Progression-free survival'. – we changed it into progression-free survival

Round 2

Reviewer 1 Report

Comments and Suggestions for Authors

Thanks for your revision.

I think that there are still many expressions that are hard to understand.

1. For example, you  described as follows "The number of collected lymph nodes different significantly between the type of surgical procedure (laparoscopy vs laparotomy) as significantly more lymph nodes were col lected during TLH procedure (10) vs laparotomy (6), p<0.001".

In my opinion, "the number of collected lymph nodes through TLH was significantly higher than that of laparotomy (p<0.001)" is general description.

2. It is necessary to verify that the data is correct.

For example, 122 had bilateral LN metastases in 131 patients having LNM based on table 2. Considering the median number (7) of collected LNs, the bilaterality (93%) is too high than my experience. 

In addition, as you mentioned above, TLH retrieved more LNs than laparotomy. However, in table 3, LNM was  significantly higher in laparotomy than TLH. 

These results are different from your hypothesis: the number of LNs removed correlates with rate of LNM and prognosis.

Please review the data carefully.

Comments on the Quality of English Language
This paper needs to be corrected in English, and it also needs to be corrected according to the thesis format including table format.

Author Response

Dear reviewer,

Thank you for your comments

For example, you  described as follows "The number of collected lymph nodes different significantly between the type of surgical procedure (laparoscopy vs laparotomy) as significantly more lymph nodes were col lected during TLH procedure (10) vs laparotomy (6), p<0.001".

In my opinion, "the number of collected lymph nodes through TLH was significantly higher than that of laparotomy (p<0.001)" is general description.

Thank you for your comment, we have edited this paragraph

  1. It is necessary to verify that the data is correct.

For example, 122 had bilateral LN metastases in 131 patients having LNM based on table 2. Considering the median number (7) of collected LNs, the bilaterality (93%) is too high than my experience. 

We have checked these values in basic data and the results are correct

Despite a great number of patients whom underwent lymphadenectomy during this study only 131 patients were found to have positive LNs. As reported in the pathology findings, the number of collected lymph nodes was also limited what may affect the study results.

We agree that the bilaterality of the findings seem to be very high, however due to the study’s retrospective design the results should be interpreted with caution.

In addition, as you mentioned above, TLH retrieved more LNs than laparotomy. However, in table 3, LNM was  significantly higher in laparotomy than TLH. 

The results in Table 3 correspond to the number of patients who underwent the procedure/ in whom lymph node metastasis were found etc. We have added appropriate annotations to the table. Please see the improved version

Reviewer 2 Report

Comments and Suggestions for Authors

I accept. Great job.

Author Response

Dear reviewer,

Thank you for your cooperation.

We have added some additional revisions in accordance to the comments of the other reviewer and academic editor

Please see the improved version of the manuscript